# Metabolic Syndrome and Male Fertility: Beyond Heart Consequences of a Complex Cardiometabolic Endocrinopathy

**DOI:** 10.3390/ijms23105497

**Published:** 2022-05-14

**Authors:** Gianmaria Salvio, Alessandro Ciarloni, Melissa Cutini, Nicola delli Muti, Federica Finocchi, Michele Perrone, Silvia Rossi, Giancarlo Balercia

**Affiliations:** Division of Endocrinology, Department of Clinical and Molecular Sciences (DISCLIMO), Polytechnic University of Marche, 60126 Ancona, Italy; g.salvio@pm.univpm.it (G.S.); a.ciarloni@pm.univpm.it (A.C.); melissa.cutini@ospedaliriuniti.marche.it (M.C.); nidellimuti@gmail.com (N.d.M.); f.finocchi@staff.univpm.it (F.F.); m.perrone@pm.univpm.it (M.P.); s.rossi@pm.univpm.it (S.R.)

**Keywords:** couple infertility, sterility, sperm quality, abnormal semen analysis, overweight, obesity, visceral fat, cardiovascular risk

## Abstract

Metabolic syndrome (MetS) is a highly prevalent condition among adult males, affecting up to 41% of men in Europe. It is characterized by the association of obesity, hypertension, and atherogenic dyslipidemia, which lead to premature morbidity and mortality due to cardiovascular disease (CVD). Male infertility is another common condition which accounts for about 50% of cases of couple infertility worldwide. Interestingly, male infertility and MetS shares several risk factors (e.g., smoking, ageing, physical inactivity, and excessive alcohol consumption), leading to reactive oxygen species (ROS) production and increased oxidative stress (OS), and resulting in endothelial dysfunction and altered semen quality. Thus, the present narrative review aims to discuss the pathophysiological mechanisms which link male infertility and MetS and to investigate the latest available evidence on the reproductive consequences of MetS.

## 1. Introduction

Overweight and obesity are conditions defined by excess adipose tissue and associated with premature morbidity and mortality that are highly prevalent in the general population [1]. According to the World Health Organization (WHO) definition, excess adipose tissue is established using the body mass index (BMI), which is calculated by dividing body weight in kilograms by the square of the height in meters, with normal BMI ranging from 18.5–24.9 kg/m^2^ and cut-offs for overweight and obesity starting from 25 kg/m^2^ and 30 kg/m^2^, respectively [2]. Since 1980, prevalence of overweight and obesity doubled in the American population, reaching 64.2% and 28.3%, respectively, in 2015. Similarly, in Europe, the prevalence of overweight and obesity increased to 49.6 and 19.6% [1]. The metabolic syndrome (MetS) is a clinical condition that, in addition to excess adipose tissue, also includes hypertension, insulin resistance, and dyslipidemia (Figure 1), with a prevalence of 35% in America and slightly higher in Europe (41% in men and 38% in women). According to recent reports, smoking, ageing, physical inactivity, and excessive alcohol consumption are recognized as risk factors for MetS [3]. Male infertility is another high prevalent condition that shares some of the risk factors of MetS is Males, indeed, are responsible for 50% cases of couple infertility, defined by the WHO as the absence of conception after 12 months of regular, unprotected intercourse [4], which is estimated to affect 48.5 million couples around the world [5]. In 30–50% of male infertility cases a specific cause is not identified and are therefore defined as “idiopathic”, but risk factors such as smoking, alcohol, obesity and ageing may contribute to impair semen quality [6]. Indeed, all these conditions are associated to increased oxidative stress (OS), which has been hypothesized to be involved in up to 80% of idiopathic male infertility cases [7]. The aim of this narrative review, therefore, is to discuss the current evidence on the relationship between MetS and male infertility, with particular attention to the biological mechanisms involved in OS.

## 2. Definition of Metabolic Syndrome

MetS is defined as a cluster of risk factors for cardiovascular disease (CVD) and type 2 diabetes mellitus (T2DM), which are more often observed together than alone. The first to notice this type of association was Kylin, almost a hundred years ago [8]. Since then, different diagnostic criteria and different names have been proposed for this pathological condition (Syndrome X, The Insulin Resistance Syndrome, etc.). Although different definitions of MetS exist, all of them include insulin resistance and its surrogates, such as high blood pressure, obesity, and atherogenic dyslipidemia (elevated triglyceride levels and low high-density lipoprotein -HDL- cholesterol levels) [9,10,11,12,13,14] (Table 1).

## 3. Metabolic Syndrome and Male Infertility: Pathophysiological Aspects

### 3.1. Metabolic Syndrome and Oxidative Stress

As mentioned above, the elements that characterize MetS are insulin resistance, hypertension, dyslipidemia, and obesity, and they are all associated increased OS [15]. Indeed, OS derives from an imbalance between the production and inactivation of reactive oxygen species (ROS), and both these mechanisms are involved in MetS. In particular, it has been shown that obese and MetS subject have lower levels of antioxidant (AOX) molecules, such as Vitamin C and tocopherol, and reduced activity of enzymes responsible for neutralizing ROS, such as superoxide dismutase [16,17], whereas the AOX administration positively influence OS biomarker levels in MetS patients [18]. The increased production of ROS, on the other hand, results from increased enzymatic activity at both the cytosolic and mitochondrial levels. In this purpose, mitochondria are the main source of ROS in most of the mammalian cells. The excess of nutrients in adipocytes, as in case of hypercaloric diets, leads to increased mitochondrial fatty acid oxidation, resulting in reduced nicotinamide adenine dinucleotide (NADH) and reduced flavin adenine dinucleotide (FADH2) production. In the mitochondrial electron transport chain, NADH and FADH2 donate electrons to the complexes I and II of the mitochondrial electron transport chain, and the latter donate electrons to Coenzyme Q10 (CoQ10) and complex III. Electrons leaking from the electron transport chain react to oxygen to generate ROS (namely superoxide radicals and hydrogen peroxide) which can damage membranes, proteins, enzymes and deoxyribonucleic acid (DNA) [19]. The interaction between ROS and mitochondria leads to mitochondrial dysfunction, which is characterized by decrease in number and altered activity of oxidative proteins, resulting in further generation of ROS, diminished ATP production and reduced energy metabolism. The imbalance between energy production and utilization has been hypothesized to be the basis of the reduced cellular metabolism that underlies the development of insulin resistance typical of MetS [20]. Other mechanisms which contribute to generate OS in MetS are represented by hyperglycemia, increased levels of advanced glycation end products, free fatty acids, and angiotensin II, together with dysregulated production of adipokines and by a state of chronic low-grade inflammation [21,22,23]. As result, there is an accumulation of oxidized products in carbohydrate, lipid and protein molecules with consequences on the respective biological functions and the impairment of intracellular pathways [24,25]. A clear example is provided by the increased oxidized low-density lipoprotein (LDL) levels in men with obesity and MetS [26]. Oxidized LDL, indeed, plays a central role in atherosclerosis and in turn contributes to an increase in the formation of ROS and to perpetuate a pro-inflammatory state [27]. Another example is given by the alteration of endothelial function. Indeed, in presence of high levels of ROS, they react with NO, the main mediator of vasodilation, to form peroxynitrite, which produces direct damage as a radical, and also renders NO unavailable to mediate its physiologic functions [28]. Taken together, this evidence supports the key role of OS in MetS. However, there are still many aspects to be clarified, especially regarding the cause-effect relationship between the two.

#### 3.1.1. Seminal Oxidative Stress

ROS are highly reactive oxidizing agents; these include hydrogen peroxide (H_2_O_2_), superoxide anion (O_2_-), nitric oxide (NO) and hydroxyl radical. Normally, in the seminal fluid, the quantity of ROS is counterbalanced by AOX substances. ROS play, in fact, an important role in the mediation of reactions such as capacitation, hyperactivation and acrosomal reaction [29]. When there is an imbalance between ROS production and the neutralizing activity of the AOX system, the condition of OS arises.

Seminal ROS can be produced both endogenously and exogenously [30]. Most ROS are produced endogenously by both leukocytes and mitochondria of immature sperm. This happens because mitochondria generate energy through oxidative phosphorylation, in particular redox reactions are coupled to the transfer of protons (H^+^) across the mitochondrial membrane to produce ATP [13]. During oxidative phosphorylation, in addition to water, a small percentage of O_2_^−^ is also synthesized [31]. On the plasma membrane of the sperm, the enzyme nicotinamide adenine dinucleotide phosphate (NADPH) oxidase catalyzes the synthesis of superoxide by transferring an electron to oxygen from NADPH [14]. In addition, leukocytes, mainly polymorphonuclear neutrophils (PMN) or granulocytes, can be activated in the presence of a chronic infection or inflammation caused by infection of the male accessory glands [32]. The activation of leukocytes determines an increase in the production of NADPH with a consequent increase in the concentration of superoxide anion and therefore OS [33].

As far as exogenous sourced of ROS are concerned, incorrect social behaviors (alcohol abuse, smoking and increased BMI), exposure to pollutants or toxic substances and the abuse of drugs and/or medications are all associated to increased OS. Of course, stress and aging also contribute to the increase in ROS in the semen. From a pathophysiological point of view, all these elements act by determining inflammation on the seminal tract. Inflammation amplifies OS by generating highly reactive substances, and ROS, in turn, attract and activate additional immune cells. Obesity, in particular, causes systemic inflammation sustained by T helper 1 lymphocytes that produce cytokines and pro-inflammatory mediators which are associated with suppression of the hypothalamus-pituitary-gonadal axis and increased intratesticular OS [34].

The mechanisms linking MetS and seminal OS are summarized in Figure 2.

#### 3.1.2. Methods to Measure Oxidative Stress

Seminal OS can be measured by both direct and indirect tests. Direct tests measure the concentration of oxidant molecules while indirect tests measure the concentration of AOXs or analyzes the ROS—induced damage on cellular components, such as DNA, proteins and lipids. Unfortunately, there is no gold standard for the evaluation of seminal OS, because each technique has its advantages and disadvantages. The most common direct tests are represented by chemiluminescence method, nitro blue tetrazolium (NBT) assay, cytochrome C reduction test, electron spin resonance technique, use of fluorescein probe, and oxidation-reduction potential (ORP). The most modern methods are luminol and ORP. The first technique is based on the chemiluminescent response of luminol when it reacts with a free radical. This response can be measured by luminometry and the number of relative light units (RLU) per million sperm calculated. One of the main problems with this technique is that luminol is sensitive to pH, temperature changes and interference from such as ascorbic acid (decreased signal) or thiol-containing molecules (which increase results) and levels of other proteins present. The ORP method is based on the direct measurement of the REDOX balance of a sample by electrochemical means. As a supplementary measure to the combined sample that it requires minimal manipulation and therefore is quite standardizable, it is currently a topic for a lot of subfertility research. There is currently only one machine on the market, the MiOXSYS (e Male Infertility Oxidative Stress System), based on a galvanostatic measure of the electron movement and provides information on the complete oxidation-reduction activity within a given sample [35]. A recent study by Douglas et al. states that measuring ORP with this device has been shown to be predictive of both poor sperm quality and male infertility [36].

Differently from direct tests, indirect tests are used to evaluate ROS released either by leukocytes or by pro-inflammatory cytokines and other immunological components. The Granulocyte elastase enzyme immunoassay and Myeloperoxidase test (Endtz test) are used to evaluate the ROS released by leukocytes [35]. Cytokine and immunological factors can be measured using Bio-Plex or enzyme-linked immunosorbent assay to quantify such molecules in the semen [37]. Although pro-inflammatory cytokines increase lipid peroxidation of the sperm membranes and contribute to OS, they are intermediate rather than causative factors of OS [38].

Recently, the 6th edition of the WHO manual for semen analysis has been published and both seminal OS and ROS have been introduced as part of semen analysis [39]. However, the manual includes only a brief description of the procedures and problems related to luminol and ORP which, as already mentioned, are the most up-to-date techniques. As evidenced by a recent analysis by Boitrelle et al., the new edition of the manual lacks the recent bibliography highlighting the predictive power of seminal OS determined by ORP [40]. Furthermore, according to the WHO, tests to evaluate OS are considered extremely specialized and used mainly in the research field.

#### 3.1.3. Effects of Oxidative Stress on Sperm Quality and the Treatment with Antioxidant Molecules

OS has been associated with male infertility and alteration of seminal sperm parameters, in particular concentration and motility [41,42]. Indeed, spermatozoa are very susceptible to ROS effects because their membranes are rich in polyunsaturated fatty acids, and this makes spermatozoa more susceptible to lipid peroxidation [33]. In addition, their cytoplasm contains very low levels of scavenging molecules [43]. But, as mentioned above, physiological levels of ROS are needed for processes like capacitation, acrosome reaction, chromatin condensation, cell signaling and sperm motility [7]. In addition, ROS production is a physiological consequence of aerobic cells’ metabolism that occurs in mitochondria, from which hydroxyl radical are released as obligatory end products [44]. Notably, OS is the major cause of sperm DNA fragmentation (SDF), and this one is an important factor in the etiology of male infertility [45]. According to this, some Authors proposed the term “Male Oxidative Stress Infertility” (MOSI), to describe infertile men with abnormal semen characteristics and high OS, reclassifying, in this way, subjects who were considered to suffer from idiopathic male infertility [7].

Based on these assumptions, it would be reasonable to consider the use of AOXs with the objective of improving the semen quality in this type of patients. In this purpose, there are several studies and clinical trials focused in the effect of AOXs on sperm [46,47,48]. We recently conducted a systematic review focused on CoQ10 on its effects on semen quality [49] but evidence on vitamin C, vitamin E, selenium and zinc as lowering OS agents for male infertility treatment also exist [50,51,52,53]. Recent studies have also investigated the promising effects of MYO-inositol, in particular for patients undergoing medically assisted reproductive procedures [54,55]. However, the efficacy of AOXs in the treatment of male infertility is still debated. Indeed, the latest European Association of Urology (EAU) guidelines state that “No clear recommendation can be made for the treatment of patients with idiopathic infertility using AOXs, although AOX use may improve semen parameters” due to poor quality of the available evidence [56]. Similarly, American Urology Association / American Society for Reproductive Medicine (AUA/ASRM) guidelines state “Clinicians should counsel patients that the benefits of supplements (e.g., AOXs, vitamins) are of questionable clinical utility in treating male infertility. Existing data are inadequate to provide recommendation for specific agents to use for this purpose” [57]. However, the lack of efficacy of AOXs in the treatment of male infertility could in part result from the failure to select subjects with high OS in available trials. Indeed, the recent position statement from the Italian Society of Andrology and Sexual Medicine (SIAMS) on the use of nutraceuticals in male sexual and reproductive disturbances suggest considering the use of AOXs in selected patients, i.e., those with idiopathic infertility in the presence of documented abnormal sperm parameters and altered SDF [58].

Notably, although considered safe and without major side effects, it must be taken into account that the excessive administration of AOXs can lead to the appearance of reductive stress, which in turn has negative effects on seminal quality [59]. Moreover, a recent review highlighted a statistically significantly increased risk of nausea (Odds Ratio: 2.16, 95% CI, *p* = 0.036), headache (Odds Ratio: 3.05, 95% CI, *p* = 0.001), and dyspepsia (Odds Ratio: 4.12, 95% CI, *p* = 0.009) in patients treated with AOXs compared to placebo [60].

In conclusion, further studies and randomized controlled trials (RCTs) are needed to define the correct dosage and type of nutraceutical products and role of AOXs in infertility, especially for idiopathic forms.

### 3.2. Metabolic Syndrome and Semen Quality

#### 3.2.1. Standard Semen Parameters

The set of metabolic disorders represented by abdominal obesity, dyslipidemia, hypertension and insulin resistance, could be involved in worsening of sperm parameters and in the pathogenesis of male infertility [61]. We have investigated the possible associations between MetS and seminal parameters and contrasting data have been reported. In fact, we have found possible associations between MetS and a lower semen volume [62,63,64], lower sperm concentration [62,63,65], lower sperm motility [63,65,66,67], altered sperm morphology [62,67,68] and vitality [65,69]. All data found are summarize in Table 2. MetS may lead to reduced male fertility via multiple mechanisms including altered hormonal profiles, epigenetic changes, increased testicular temperature, inflammation, and OS [70,71,72,73]. In addition, reduced bioavailability of NO due to high ROS levels causes vasoconstriction, thrombosis, inflammation and vascular hypertrophy, which together lead to endothelial dysfunction [74], with possible consequences also at gonadal level.

#### 3.2.2. Sperm DNA Fragmentation

In last years, interest in understanding the role of SDF in male infertility and its implications on reproduction has grown considerably. In fact, as human spermatozoa are highly sensitive to OS, sperm plasma membrane damage and nuclear or mitochondrial DNA fragmentation can occur in response to ROS [7]. Furthermore, a known consequence of MetS is T2DM that has increasingly been associated with male infertility, and complex and multifactorial factors are involved. In this purpose, Palmer et al. found a positive correlation between glycaemia and SDF, with a negative correlation with normal morphological spermatozoa, regardless of adiposity, in mice fed with a high fat diet [76]. According with this animal model, Leisegang et al. found higher SDF in men with MetS in two different works [63,65]. Conversely, Elfassy et al. reported no difference for percentage of DNA fragmentation between men with or without MetS [75].

#### 3.2.3. Mitochondrial Membrane Potential

Worsening of semen quality with low sperm concentration and motility, abnormal morphology, mitochondrial DNA damage, nuclear DNA damage and increased seminal plasma abnormalities have been reported in patients with Mets [77]. Sperm vitality can be easily assessed by measuring the inner mitochondrial membrane potential (MMP) in sperm cells [78]. The latter reflects the energy status of the mitochondria and is directly associated with the motility of spermatozoa [79]. Leisegang et al. found that percentage of spermatozoa with damaged MMP was significantly increased in the MetS group and reported a positive correlation between both C-reactive protein (CRP) and MMP and sperm concentration, motility, and vitality [61,63]. We could explain these evidences with the higher serum levels of CRP, inflammatory cytokines [80] and ROS [81], observed in patients with MetS that provoke damage to mitochondrial function and spermatozoa DNA [82,83].

### 3.3. Metabolic Syndrome and Reproductive Hormones

The link between hormones and MetS is complex and bi-directional, since the clinical features associated with this condition appear to be conditioned by numerous hormonal stimuli and, in turn, can influence the functioning of numerous glands in a complex two-way network. During the last years, the traditional view of adipose tissue as an energy storage tissue has been replaced by the concept of adipose tissue as an endocrine organ, able to express and secrete a large variety of bioactive peptides, including leptin and cytokines (or “adipokines”) [84]. In addition, adipose tissue is capable to aromatize androgens to estrogens, leading to lower levels of circulating testosterone due to its increased aromatization to estradiol in obese men, and hyperestrogenemia further inhibits testosterone production by negative feedback on the hypothalamus-pituitary-testicular axis [85]. According to the results of two recent meta-analysis, MetS is associated to significantly lower levels of follicle-stimulating hormone (FSH), testosterone, and inhibin B [61,86]. Since FSH is produced by the pituitary and inhibin B by testicular Sertoli cells, these data confirm the association between MetS and hypothalamus-pituitary-gonadal axis impairment at different levels. At the same time, hypogonadism per se can promote the development of MetS. Indeed, albeit the effects of androgen deprivation therapy on BMI are modest (0.65 kg/m^2^ after 12 months), it is associated to profound effects on body composition, leading to a 30% increase in insulin resistance [87]. Conversely, testosterone replacement therapy in hypogonadal man decreases body weight and insulin resistance and improve glycemic control and waist-to-hip ratio [88]. Similarly, prevalence of MetS is more than 2-fold increased in hypopituitary patients, especially when adult growth hormone deficiency (GHD) is present, and growth hormone replacement treatment has shown to improve the metabolic profile of these patients [89].

Concerning thyroid function, recent data suggest that thyroid disfunction may be associated with male fertility issues. Hyperthyroidism, indeed, seems to be related to reduced sperm mitochondrial activity, altered AOX defense, and delayed spermatogenesis, whereas hypothyroidism leads to reduced sperm vitality and delayed sperm transit through the epididymis [90]. To confirm this, thyroid dysfunction seems to be more frequent in subjects with altered seminal quality than in the general population [91]. Therefore, evaluation of thyroid function should be considered when assessing the hormonal balance of the infertile male. Conversely, the association between MetS and abnormality of thyroid hormones remains under debate. In particular, despite general consensus on the relationship of thyroid hormones and BMI in overt hypothyroidism exists [92], data on subclinical hypothyroidism (SCH) are controversial. According to the results of a recent cross-sectional study, total cholesterol and LDL levels are significantly higher in patients with SCH [93]. In addition, a recent meta-analysis has shown a significant association between hypertension and SCH, especially in middle-aged women [94]. Conversely, other authors have reported no association between thyroid hormone levels and insulin resistance indices (namely fasting glycemia, insulin and e homeostatic model assessment of Insulin Resistance, HOMA-IR) in women with SCH [95]. Unfortunately, despite the relationship between SCH and CVD have been supported by several studies [96], two recent randomised controlled trials have shown that treatment with levothyroxine seems to be ineffective in reducing the risk of cardiovascular outcomes in older patients with SCH [97]. Due to the scarcity of available data, the European Thyroid Association suggest levothyroxine treatment in SCH patients only if symptoms consistent with hypothyroidism are present [98], whereas the American Thyroid Association guidelines suggest considering treatment also in patients at increased CVD risk [99].

Endogenous hypercortisolism (Cushing’s syndrome, CS) presents several clinical features associated to high risk of CVD, including central obesity, hypertension, glucose intolerance, dyslipidemia, resembling polycystic ovary syndrome (PCOS) in women and MetS in men [100]. Chronic glucocorticoids exposure, indeed, lead to the fat redistribution typical of CS, with preferential visceral fat accumulation [101]. This effect may be observed also in functional hypercortisolism (also referred to as “Pseudo-Cushing”), a potentially reversible condition associated to several clinical states such as diabetes mellitus, obesity, and depression [102]. Similarly, subclinical CS, recently renamed mild autonomous cortisol excess (MACE), a common finding in adrenal incidentalomas, is frequently associated to hypertension, dyslipidemia, diabetes mellitus, with a doubled risk of new cardiovascular events compared with subjects with nonfunctioning adenomas [103].

## 4. Metabolic Syndrome and Drugs: Fertility Issues

Since patients with MetS often take one or more medications, including antihypertensives, hypoglycemic agents, and cholesterol-lowering agents, any potential benefits or side effects of these drugs on the reproductive system must be considered.

Concerning antihypertensives, the main drugs used in the treatment of hypertension are as angiotensin converting enzyme (ACE) inhibitors and angiotensin-receptor blockers (ARBs), two different class of medications that lower blood pressure targeting the renin-angiotensin system (RAAS). RAAS is activated by the release of renin from juxtaglomerular cells of renal afferent arteriolas in case of low blood pressure. As first step, renin converts angiotensinogen in angiotensin I, which is then transformed in angiotensin II by ACE, localized mostly in the endothelial cells of the lungs. Then, Angiotensin II increases blood pressure by a direct vasoconstrictive effect and also by stimulating the adrenal secretion of Aldosterone [104]. There is evidence in literature that RAAS could be involve in male reproduction. Leal et al., indeed, using immunofluorescence, revealed the presence of angiotensin (1–7) receptor Mas in the testes of mice and rats [105]. Moreover, they found that Mas-deficient mice showed reduced testis weight and an increased number of apoptotic cells in seminiferous epithelium, with lower daily sperm production, compared with wild-type mice [105]. In addition, ACE activity has been reported to be positively associated with sperm concentration and fertility in animal models [106]. By the way, the pharmacologic inhibition of RAAS seems to have mixed effects on sperm quality. Bechara et al. studied the effects of the therapy with Enalapril on testes volume and sperm production of spontaneous hypertensive rats, reporting increased testicular vascular volumetric density, volumetric density of seminiferous epithelium and sperm production in the treated group compared with the non-treated group [107]. Conversely, Altintas Aykan at al. reported a significant reduction in sperm motility in normotensive and dexamethasone-induced hypertensive adult male rats treated with RAAS inhibitors (sacubitril/valsartan and ramipril) [108], that could be explained by the presence of an Angiotensin receptors on the tail of rat’s spermatozoa [109].

Calcium-channel blockers (CCB) are a group of antihypertensive drugs that targets calcium channels located at the level of the plasma membrane and activated by cell depolarization. In resting cells, those channels are usually closed; opening those channels allows the entry of calcium ions according with their electrochemical gradient. CCBs exert their antihypertensive effect inhibiting external calcium-evocated contraction in depolarized arteries [110]. The anti-fertility effects of CCBs have been evaluated both in vitro and in vivo [111]. In this purpose, Murakinyo et al. studied the effects of these drugs on sperm parameters and sexual hormones levels in male rats. Nifedipine, Verapamil and Diltiazem significantly reduced sperm count and motility and epididymal weight and those parameters were restored after 30 days of drug withdrawal. Those effects were not associated with the inhibition of pituitary-gonadal axis and were probably caused by the decreased levels of intracellular calcium levels and the consequent reduction of sperm motility [112]. Subsequently, the same authors evaluated the effects of these three drugs on the oxidative balance and the functionality of rat’s spermatozoa, finding increased OS stress and inhibition of the acrosomal reaction in the treated animals [113]. In another animal study, Aprioku et al. did not find any alteration in sperm parameters in a group of guinea pigs treated with Nifedipine alone, but it could be partly due the short duration of treatment (only 14 days versus 30 days of Murakinyo et al.) [114].

Beta blockers are a group of drugs which exert their action on different beta-adrenergic receptors, blocking the action of endogenous catecholamines, and consequently causing an antihypertensive effect through the inhibition of the sympathetic nervous system [115]. Beta-adrenoreceptors antagonist have the potential to reduce sperm motility, as previously reported by Hong et al., who also noticed that the more those drugs are lipid soluble the lower concentrations are required to obtain this effect, probably those drugs could exert this effect by stabilizing cellular membranes [116]. In addition, it should be noticed that those drugs may also have an indirect effect on male fertility due to their association with erectile disfunction [117].

Concerning antidiabetic drugs, metformin and glucagon-like peptide 1 (GLP1) agonists may have positive effects on semen quality. Metformin exerts its glycemia-lowering effect through the reduction of hepatic gluconeogenesis, but its mechanism of action is still not completely understood [118]. Yan et al. reported that metformin has a positive effect on reproduction in obese rats, reducing the oxidative damage on the blood/testis barrier and the ectopic lipid deposition in testis [119]. In addition, the administration of metformin showed a protective effect on semen quality and testis structure was and increased levels of testosterone in obese rats [120]. This could be not only due to the metabolic action of metformin, but also to its effect on OS. Indeed, Alves at al. studied rat’s Sertoli cells cultured with and without metformin and they found out that it induces alanine production, which maintains NADH/NAD+ equilibrium with AOX effect [121]. As far as GLP1 agonists are concerned, they may have a positive effect on sperm quality, as suggested by the recent finding of GLP1 receptors on spermatozoa and Sertoli cells [122], but clinical data are lacking.

Finally, regarding lipid-lowering drugs, hydroxymethylglutaryl CoA reductase inhibitors (or statins) may have a positive effect on male fertility through their AOX action [123], in particular at the testicular level [124]. Surprisingly, Pons-Rejraji et al. reported a negative effect of atorvastatin on sperm parameters in normocholesterolemic and normozoospermic patients, with significantly reduced sperm concentration, motility and morphology and altered acrosome reaction. Notably, at list one altered parameter was present in 35% of patients during the 5 months treatment and in 65% of them after 3 months of withdrawal [125].

## 5. Metabolic Syndrome and Reproductive Outcomes

Despite several studies evaluating the effects of maternal metabolic status and fertility exist, there is absolute scarcity of data on the influence of paternal MetS on reproductive outcomes. According to a retrospective report by Kasman et al. on almost one million pregnancies, the risk of pregnancy loss was significantly higher in couple with paternal MetS. In addition, the number of paternal components of MetS significantly increased the relative risk (RR) of pregnancy loss (RR 1.10 for one, RR 1.15 for two and RR 1.19 for three or more components) [126]. Similarly, Murugappan et al. reported a significant association between the number of preconception paternal MetS component and adverse maternal outcomes, including preeclampsia, eclampsia, and severe maternal morbidity [127]. Few studies evaluated the efficacy of lifestyle intervention on reproductive outcomes in women [128], and the impact of MetS treatment on semen quality is not clear [129,130]. These data underscore the absolute need for studies in large population samples to fully understand the role of paternal MetS as a cause of couple infertility and whether treatment of its components, in addition to being desirable for improved overall health, may influence reproductive outcomes.

## 6. Conclusions

MetS and male infertility are undoubtedly linked by a relationship that includes numerous pathogenetic mechanisms. Among these, OS seems to play a primary role, but metabolic and hormonal alterations and drug interactions contribute to the impaired reproductive outcome of these patients. Clinicians should be aware of the important repercussions of MetS on semen quality and should make every effort to raise the consciousness of patients presenting with infertility problems on this topic. Large, well-designed studies are needed to determine whether correction of the pathologies that contribute to the definition of MetS may play a role in improving reproductive chances.

## Figures and Tables

**Figure 1 ijms-23-05497-f001:**
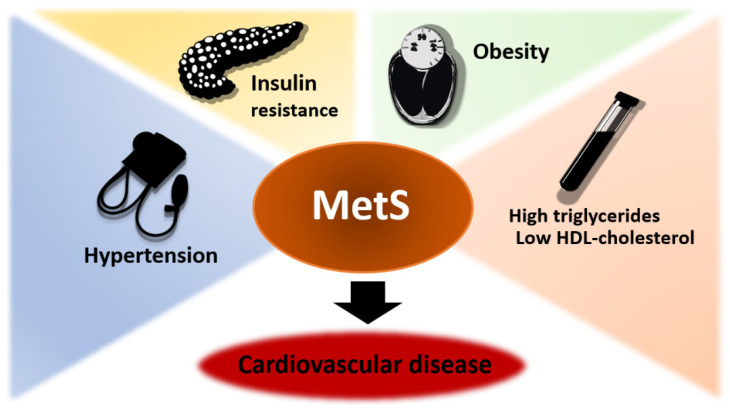
Metabolic syndrome (MetS): key-factors and cardiovascular risk.

**Figure 2 ijms-23-05497-f002:**
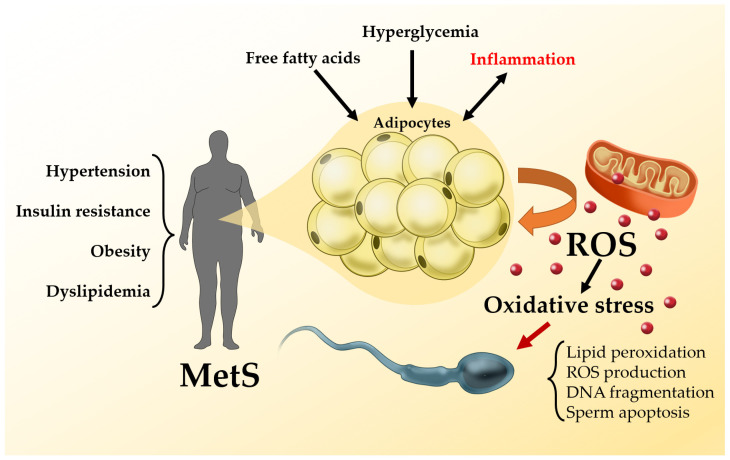
Metabolic syndrome (MetS) and oxidative stress (OS): hypertension, insulin resistance, obesity and dyslipidemia are the key-elements of MetS and they are all associated with increased OS. The latter derives from an imbalance between production an inactivation of reactive species of oxygen (ROS), which mainly derive from hyperreactive elements leaking from the mitochondrial electron transport chain. Free fatty acids, hyperglycemia and systemic inflammation contribute to sustain ROS production that in turn causes mitochondrial dysfunction and impairs cellular metabolism. High levels of intratesticular ROS, as observed in patients with MetS, may affect semen quality by causing structural and functional damage in the spermatozoa.

**Table 1 ijms-23-05497-t001:** Definition of Mets according to different scientific societies.

	WHO	EGSIR	ATP III	AACE	IDF	AHA/NHLBI	AHA/NHLBI + IDF
Definition	Insulin resistance + any other two components	Plasma insulin concentration > 75th percentile of nondiabetic patients + any of two components	Any of three out of five components	Insulin resistance + at least two other components	Central Obesity + at least two other components	Any of three out five components	Any of three out five components
Obesity	Waist/hip ratio > 0.9 in males and >0.85 in females or BMI > 30 kg/m^2^	Waist circumference ≥ 94 cm in males and ≥80 cm in females	Waist circumference >102 cm in males and >80 cm in females	BMI > 25 kg/m^2^	Obesity defined as waist circumference with ethnicity specific values or BMI >30 or kg/m^2^	Waist circumference > 40 inches in males and >35 inches in females	Raised waist circumference (Population- and country-specific definitions)
HDL	<35 mg/dL: males <39 mg/dL: females	<39 mg/dL: males and females, or specific treatment	<40 mg/dL: males <50 mg/dL: females, or specific treatment	<40 mg/dL: males <50 mg/dL: females	<40 mg/dL: males <50 mg/dL: females, or specific treatment	<40 mg/dL: males <50 mg/dL: females, or specific treatment	<40 mg/dL: males <50 mg/dL: females, or specific treatment
TG	≥150 mg/dL	≥150 mg/dL or specific treatment	≥150 mg/dL or specific treatment	≥150 mg/dL	≥150 mg/dL or specific treatment	≥150 mg/dL or specific treatment	≥150 mg/dL or specific treatment
Hyperglycemia	Impaired glucose tolerance, impaired fasting glucose, or lowered insulin sensitivity	Fasting plasma glucose > 110 mg/dL	Fasting plasma glucose > 110 mg/dL or specific treatment	Impaired glucose tolerance or impaired fasting glucose (but not diabetes)	Fasting plasma glucose > 100 mg/dL or previously diagnosed type 2 diabetes	Fasting plasma glucose > 100 mg/dL or specific treatment	Fasting plasma glucose > 100 mg/dL or specific treatment
Blood Pressure	≥140/90 mm Hg	≥140/90 mm Hg or specific treatment	SBP ≥ 130 or DBP ≥ 85 mm Hg or specific treatment	≥130/85 mm Hg	SBP ≥ 130 or DBP ≥ 85 mm Hg or specific treatment	≥130/85 mm Hg or specific treatment	≥130/85 mm Hg or specific treatment
	Urine albumin ≥ 20 μg/min or albumin: creatinine ratio ≥ 30 mg/g						

AACE, American Association of Clinical Endocrinologists; AHA/NHLBI, American Heart Association/National Heart, Lung, and Blood Institute; ATP, Adult Treatment Panel; BMI, body mass index; BP, blood pressure; DBP, diastolic blood pressure; EGSIR, European Group for the Study of Insulin Resistance; HDL, high-density lipoprotein; IDF, International Diabetes Federation; MetS, metabolic syndrome; SBP, systolic blood pressure; TG, triglycerides; WHO, World Health Organization.

**Table 2 ijms-23-05497-t002:** Standard semen parameters: comparison of MetS versus no-MetS men.

Reference	Semen Volume	Sperm Concentration	Sperm Motility	Sperm Morphology	Sperm Vitality	SDF	MMP
Ventimiglia et al. [62]	↓	↓	=	↓	=	NE	NE
Leisegang et al. [63]	↓	↓	↓ Total and progressive motility	NE	NE	↑	↓
Saikia et al. [64]	↓	Lower total count	↓ Total and progressive motility	=	NE	NE	NE
Leisegang et al. [65]	=	↓	↓ Only progressive motility	NE	↓	↑	↓
Ozturk et al. [66]	NE	Lower total count	↓	=	NE	NE	NE
Chen et al. [67]	=	NE	↓	↓	↓	NE	NE
Lotti et al. [68]	=	=	=	↓	NE	NE	NE
Elsamanoudy et al. [69]	=	NE	↓	↓	↓	NE	NE
Elfassy et al. [75]	=	↑	↓	=	↓	=	NE

* MMP, Mitochondrial membrane potential; NE, not evaluated; SDF, Sperm DNA Fragmentation; ↓, decreased; ↑, increased; =, unchanged.

## Data Availability

Not applicable.

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
