# Peer review of "Metabolic Syndrome and Male Fertility: Beyond Heart Consequences of a Complex Cardiometabolic Endocrinopathy"

_ijms, 2022, doi:10.3390/ijms23105497_

Round 1

Reviewer 1 Report

The review article by Salvio et al., aims to discuss the pathophysiological mechanisms, which link male infertility and MetS, and to investigate the latest available evidence on the reproductive consequences of MetS.

In general, the article is well written and has significance data in this field. I have the following comments:

  • A representative figure of the metabolic syndrome and oxidative stress is needed.
  • There is an error in the numbering of the sections of the article.
  • The role of mitochondria in the development of metabolic syndrome and its effects on sperm quality should be expanded.
  • It would be good to define and expand which are the main sources of ROS and their effects on fertility.

Author Response

The review article by Salvio et al., aims to discuss the pathophysiological mechanisms, which link male infertility and MetS, and to investigate the latest available evidence on the reproductive consequences of MetS.

In general, the article is well written and has significance data in this field.

Thank you for taking the time to review our work and for the valuable suggestions.

I have the following comments:

  • A representative figure of the metabolic syndrome and oxidative stress is needed.
  • Thanks for the suggestions. We added Figure 2.

  • There is an error in the numbering of the sections of the article.
  • Thank you for the notification, we have corrected the numbering of the sections.

  • The role of mitochondria in the development of metabolic syndrome and its effects on sperm quality should be expanded.
  • We do appreciate this suggestion, and we expanded the role of mitochondria in the relative sections.

  • It would be good to define and expand which are the main sources of ROS and their effects on fertility.
  • Thank you for this valuable suggestion.

Reviewer 2 Report

The topic of the manuscript is interesting and addresses problems of increasing importance.

The abstract does not present appropriately the content of the manuscript. For instance it starts with the size of the problem of infertility. That topic however is omitted in the manuscript altogether.

Some remarks on the structure of the review:

  1. It seems that the structure lacks an internal logic.
  2. The title includes the expression “cardiometabolic endocrinopathy”. Not much is said of the cardio- part in the text.
  3. The extensive comment on the different definitions of the MetS is not necessary if the main point is the male infertility.
  4. Page 7, paragraph 2: thyroid dysfunction has been included into the Metabolic syndrome and reproductive hormones section. The paragraph has no much in common with the topic of the manuscript. There are recent data demonstrating that thyroid dysfunction in the males might be associated with fertility issues. I would suggest that the paragraph is either rewritten in the light of male fertility and MetS or removed altogether.

The references

Some of the references are redundant – two or more citations of papers by one team with no real difference between the papers are included: for instance 23 and 24; 30 and 31; 21 and 78 (virtually identical review papers); 65 and 67 probably describe the same cohort. Please revise the references and remove those which do not contribute to the text.

Reference 3 is not cited very properly.

Page 2, paragraph 1: What is meant by "central core"?

Author Response

The topic of the manuscript is interesting and addresses problems of increasing importance.

  • The abstract does not present appropriately the content of the manuscript. For instance it starts with the size of the problem of infertility. That topic however is omitted in the manuscript altogether.
  • Thanks for your comment. We revised the abstract to make it more appropriate and give greater relevance to the MetS.

Some remarks on the structure of the review:

  • It seems that the structure lacks an internal logic.
  • Taking this suggestion into account, we have modified the structure of the review as follows:
  1. Introduction
  2. Definition of MetS
  3. METABOLIC SYNDROME AND MALE INFERTILITY: pathophysiological aspects (oxidative stress, semen quality, hormones)
  4. METABOLIC SYNDROME AND DRUGS: FERTILITY ISSUES
  5. METABOLIC SYNDROME AND REPRODUCTIVE OUTCOMES
  6. Conclusions

  • The title includes the expression “cardiometabolic endocrinopathy”. Not much is said of the cardio- part in the text.
  • The authors' idea was to emphasize the fact that the metabolic syndrome has repercussions on fertility, in addition to being considered a condition of high cardiovascular risk. The juxtaposition of the term cardio- and endocrinopathy was precisely intended to synthetically highlight these two aspects of the metabolic syndrome. A possible alternative could be: "Metabolic syndrome and male fertility: beyond heart consequences of a complex cardiometabolic endocrinopathy".

  • The extensive comment on the different definitions of the MetS is not necessary if the main point is the male infertility.
  • We agree, we shortened the definition paragraph and kept table 1.

  • Page 7, paragraph 2: thyroid dysfunction has been included into the Metabolic syndrome and reproductive hormones section. The paragraph has no much in common with the topic of the manuscript. There are recent data demonstrating that thyroid dysfunction in the males might be associated with fertility issues. I would suggest that the paragraph is either rewritten in the light of male fertility and MetS or removed altogether.
  • We agree that the paragraph needed to be put into context. Therefore, we decided to keep it by adding a short introduction that could justify it.

The references

  • Some of the references are redundant – two or more citations of papers by one team with no real difference between the papers are included: for instance 23 and 24; 30 and 31; 21 and 78 (virtually identical review papers); 65 and 67 probably describe the same cohort. Please revise the references and remove those which do not contribute to the text.
  • Thank you, we have removed duplicate sources from those reported and checked for additional redundant sources.

  • Reference 3 is not cited very properly.
  • We agree, reference 3 has been removed.

  • Page 2, paragraph 1: What is meant by "central core"?
  • The meaning was that of "key elements". However, the sentence has been removed.

Thanks for the valuable suggestions. We have also extensively revised the article to correct any grammatical errors.

Round 2

Reviewer 1 Report

No more comments